# Building up libraries and production line for single atom catalysts with precursor-atomization strategy

Xiaohui He [1], Hao Zhang[1], Xingcong Zhang[1], Ying Zhang[1], Qian He[1], Hongyu Chen[1], Yujie Cheng[1], Mi Peng[2], Xuetao Qin[2], Hongbing Ji [1,3] & Ding Ma [2]

Having the excellent catalytic performance, single atom catalysts (SACs) arouse extensive research interest. However, the application of SACs is hindered by the lack of versatile and scalable preparation approaches. Here, we show a precursor-atomization strategy to produce SACs, involving the spray of droplets of solutions containing metal precursors onto support surface through ultrasonic atomization and the subsequent calcination. This approach is versatile to successful synthesis of a series of catalysts, including 19 SACs with different metal sites and supports and 3 derivatives of SACs (single atom alloys, double atom catalysts and bi-metallic SACs). Furthermore, it can be scaled up by a homemade production line with productivity over 1 kg day$^{-1}$, and the well-controlled catalyst uniformity is evidenced by the identical characterization results and catalytic properties in Suzuki-Miyaura cross-coupling. This strategy lays a foundation for further investigation and may accelerate the trend from basic research to industrial applications of SACs.

Single-atom catalysts (SACs), featured with spatially isolated metal atoms on underlying supports, are regarded as a class of promising materials[1], because they integrate the merits of homogeneous catalysts (high atom utilization, activity, and selectivity) and heterogeneous ones (good stability and reusability)[2], and show superior catalytic performance in practical application, including high-value chemicals production[3], energy conversion[4], and pollution eliminating[5]. However, it is still widely recognized plenty of unknowns of SACs to be explored[6,7], including discovering unique catalytic properties[8,9], understanding structure–performance relations[10,11], and realizing industrial applications[12].

The prerequisite for the further study of SACs is to develop flexible and scalable preparation methods[13], but it remains a great challenge.

First, developing a general route to fabricate various SACs with different metal sites and supports is very difficult[14–16], due to their obviously different physical/chemical properties, including but not limited to the solubility, reactivity, and thermal stability of metal precursors as well as the surface coordination environments and hydrophilic/hydrophobic properties of supports[17].

Second, the scaling-up effect seriously hampers the mass production of SACs[18,19]. Take the commonly used wet-chemistry approaches (e.g., impregnation[20], hydrothermal treatment[21], co-precipitation[22], etc.) as examples. The synthesis conditions, such as concentration, temperature, stirring rate, and solution pH, have a noticeable influence on the aggregation states of the metal species[23]. Although these conditions can be well controlled on a lab scale (so the atomically dispersed structure is obtained), it is almost impossible to realize on a plant scale on account of the inevitable uneven mass transfer and heat transfer, leading to the excessive local concentrations and then the undesired aggregation of metal species[24]. One of the solutions is to introduce continuous preparation methods (increasing the production over time) instead of the current batch-type ones, but the relative reports are very rare[25,26].

[1]Fine Chemical Industry Research Institute, School of Chemistry, Sun Yat-sen University, 510275 Guangzhou, China. [2]Beijing National Laboratory for Molecular Sciences, College of Chemistry and Molecular Engineering and College of Engineering, and BIC-ESAT, Peking University, 100871 Beijing, China. [3]Huizhou Research Institute, Sun Yat-sen Universuity, 516081 Huizhou, China. ✉e-mail: jihb@mail.sysu.edu.cn; dma@pku.edu.cn

Therefore, it is a scientific and technological imperative to develop a versatile, continuous, and straightforward strategy to fabricate SACs.

In this work, we report a precursor-atomization strategy for the synthesis of SACs and their derivatives, and the whole fabrication process includes only two steps: Step 1, the dilute solution of the precursors is atomized and sprayed onto the supports; Step 2, the above samples undergo heat treatment to decompose the precursors, and the corresponding SACs are obtained. It is worth noting that according to this simple method we can build up SACs libraries for apparently various different metal-support combinations (as many as 19 SACs and 3 derivatives) without any special precautions and the productivity can reach 1 kg day⁻¹ by a homemade production line.

## Results

### Synthesis of Pd SACs with the precursor-atomization strategy

The homemade equipment was shown in Supplementary Fig. 1, which was composed of an ultrasonic atomizer (to atomize the precursor solution), three infrared lamps and a heating plate (to remove the solvent quickly), and a plastic dome shield and an iron basin (to confine the atomization in a relatively enclosed space and facilitate the recycling of the unemployed precursors, details see the "Methods" section).

Here we take $Pd_1/FeO_x$ as an example to show the synthesis process (Fig. 1a). The aqueous solution of tetraamminepalladium nitrate ($[Pd(NH_3)_4](NO_3)_2$, 2.45 mmol L⁻¹) was atomized into very small drops of ~40 μm³ in volume by an ultrasonic atomizer[27,28] and sprayed at a rate of ~40 mL h⁻¹ onto $FeO_x$, which was spread out evenly on the iron basin. The infrared lamps and the heating plate were employed to remove the moisture quickly by heating the $FeO_x$ powders to over 60 °C. After being sprayed, the sample was carefully collected and calcined at 400 °C in a tube furnace[29,30].

To our great delight, there were no detectable Pd nanoparticles in the transmission electron microscopy (TEM) and scanning transmission electron microscopy (STEM) images of $Pd_1/FeO_x$ (Fig. 1b and Supplementary Fig. 2) and the corresponding element mapping results

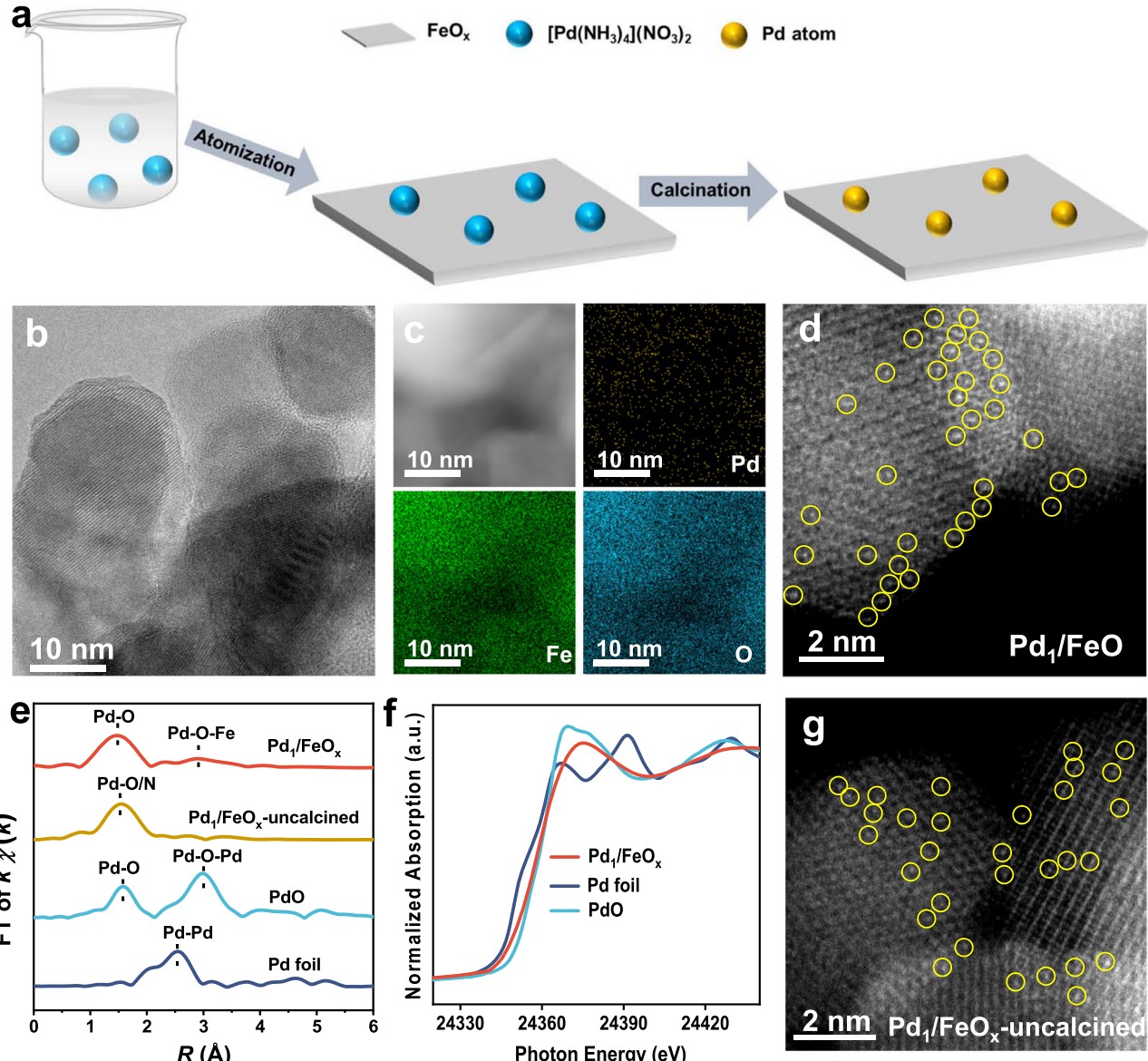

**Fig. 1 | Preparation and structural characterization of $Pd_1/FeO_x$. a** The scheme for the preparation of $Pd_1/FeO_x$ with the precursor-atomization strategy. **b** TEM image of $Pd_1/FeO_x$. Scale bar, 10 nm. **c** Element mapping of $Pd_1/FeO_x$. **d** AC HAADF-STEM image of $Pd_1/FeO_x$. Pd atoms were highlighted by yellow circles. Scale bar, 2 nm. **e** The $k^3$-weighted FT spectra of Pd K-edge EXAFS for $Pd_1/FeO_x$, $Pd_1/FeO_x$-uncalcined, PdO, and Pd foil. **f** Pd K-edge XANES for $Pd_1/FeO_x$, PdO, and Pd foil. **g** AC HAADF-STEM image of $Pd_1/FeO_x$-uncalcined.

suggested that Pd species were uniformly distributed over $Pd_1/FeO_x$ (Fig. 1c). Furthermore, from the aberration-corrected high-angle annular dark-field scanning transmission electron microscopy (AC HAADF-STEM, Fig. 1d) image, isolated Pd atoms were clearly observed as bright dots. The inductively coupled plasma optical emission spectrometry (ICP-OES) and Brunauer–Emmet–Teller (BET) measurement revealed that the Pd loading and specific surface area of $Pd_1/FeO_x$ was 0.27 wt% and 41.6 $m^2 g^{-1}$, respectively (Supplementary Table 1). The X-ray diffraction (XRD, Supplementary Fig. 2) pattern of $Pd_1/FeO_x$ showed similar peaks as $Fe_2O_3$ (PDF#87-1165), without the Pd NPs or PdO peaks, indicating the well-dispersed Pd species in $FeO_x$. There were two notable peaks at 1.5 and 2.9 Å (Fig. 1e), which can be ascribed to the Pd–O and Pd–O–Fe scattering paths, respectively, in agreement with the previous report[31]. It should be noted that the peak at 2.9 Å cannot be ascribed to the Pd–Pd scattering path (Pd foil, 2.5 Å) and Pd–O–Pd scattering path (PdO, 3.0 Å), which was further confirmed by the EXAFS fitting results of $Pd_1/FeO_x$ that these fitting results were in good agreement with the original curves (Supplementary Fig. 3 and Supplementary Table 2). Furthermore, the quantitative structural parameters results (Supplementary Table 2) of $Pd_1/FeO_x$ indicated that Pd atom was connected with four O atoms with a mean bond length of about 2.01 Å[31]. The wavelet transforms (WT) plot (Supplementary Fig. 4) of $Pd_1/FeO_x$ showed the maximum peak at ~6.2 $Å^{-1}$, corresponding to the Pd–O scattering path by comparing Pd foil with the intensity maxima of Pd–Pd at ~10.2 $Å^{-1}$[31], which agreed with the EXAFS results in R space. These results further confirmed the dominant presence of atomically dispersed Pd species. In addition, as shown in X-ray absorption near edge structure (XANES) spectra (Fig. 1f), the absorption edge positions of $Pd_1/FeO_x$ were situated between those of Pd foil and PdO, indicative of the positively charged Pd species. All these characterization results corroborated the successful synthesis of Pd SACs ($Pd_1/FeO_x$) via the precursor-atomization strategy.

In order to understand the formation process of $Pd_1/FeO_x$, the samples ($Pd_1/FeO_x$-uncalcined), which were after being sprayed but before being calcined, were also characterized. The visible Pd atoms in AC HAADF image (Fig. 1g) and the absence of Pd–Pd bonds in EXAFS results (Fig. 1e) co-indicated that the Pd species were spatially isolated, demonstrating the molecularly dispersed states of the precursors ($[Pd(NH_3)_4](NO_3)_2$). Considering the complete decomposition temperature of $[Pd(NH_3)_4](NO_3)_2$ was about 270 °C (Supplementary Fig. 5), the samples above were subsequently treated at 400 °C, which was higher than the decomposition temperature but much lower than the migration temperature of noble metal species on metal oxide supports[32,33], ensuring the transformation of molecularly dispersed $[Pd(NH_3)_4](NO_3)_2$ into atomically dispersed Pd atoms after calcination (i.e., $Pd_1/FeO_x$ was fabricated).

## The versatility of the precursor-atomization strategy

Surprisingly, the precursor-atomization strategy is of extraordinary flexibility for SACs fabrication through adjusting preparation parameters, including the type of precursors and the supports (details see Supplementary Table 3). These catalysts were characterized comprehensively by TEM, STEM, element mapping, X-ray absorption spectroscopy (XAS), XRD, ICP-OES, Elemental analysis (EA), and BET (see Supplementary Figs. 6–70 and Supplementary Tables 4–46) to determine the aggregation states and loadings of metal species as well as identify the texture properties of supports.

First, by simply replacing $[Pd(NH_3)_4](NO_3)_2$ with other precursors ($[Pt(NH_3)_4](NO_3)_2$, $Ru(NO)(NO_3)_x(OH)_y$, $Mn(NO_3)_2·4H_2O$, $Fe(NO_3)_3·9H_2O$, $Co(NO_3)_2·6H_2O$, $Ni(NO_3)_2·6H_2O$ and $Zn(NO_3)_2·6H_2O$) and replacing $FeO_x$ with other supports ($Al_2O_3$, $TiO_2$, $MnO_x$, ZnO, and nitrogen-doped carbon (N–C)), as many as 18 SACs were also prepared. As shown in Fig. 2, the atomically dispersed nature of metal species was co-demonstrated by AC HAADF-STEM images (isolated metal atoms distributed throughout the supports) and EXAFS results (no corresponding metal–metal bond detected).

Moreover, to obtain $Pd_1/Cu$ single atom alloys (SAAs)[6], the $H_2PdCl_4$ solution was sprayed on the Cu powders directly (Fig. 3a–d). As shown in Fig. 3a, there were bright dots in the catalysts of $Pd_1/Cu$, which can be ascribed to the isolated Pd atoms, and the mapping result also indicated the homogenous distribution of Pd species (Fig. 3b). The EXAFS result showed no Pd–Pd bonds, and the fitting results indicated the presence of a Pd–Cu bond with the coordination numbers of ~8, similar to the previous reports of palladium–copper SAAs[18,34] (Fig. 3c and Supplementary Fig. 62; Supplementary Table 41). The XANES and XRD results revealed that both the Pd and Cu species were primarily in the metallic states (Fig. 3d and Supplementary Fig. 60). Combining the results above, it was concluded that the $Pd_1/Cu$ SAAs were successfully fabricated. In addition, when we replaced $[Pd(NH_3)_4](NO_3)_2$ by $C_6H_{10}Cl_2Pd_2$ as the precursor, Pd DACs[7,35] were obtained ($Pd_2/FeO_x$, Fig. 3e–h), featured with 3.15 Å distance between the adjacent Pd atoms in AC HAADF-STEM images (Fig. 3e, f), and exhibited the dominant peak at 1.6 Å in EXAFS results with the absence of the signal for Pd–Pd bond (2.5 Å, Fig. 3g). XANES result (Fig. 3h) suggested the positively charged Pd species on $Pd_2/FeO_x$, similar with those on $Pd_1/FeO_x$. The results above demonstrated a unique configuration of DACs that were composed of spatially adjacent atoms without strong electronic interaction. Furthermore, when the solutions of $[Pd(NH_3)_4](NO_3)_2$ and $[Pt(NH_3)_4](NO_3)_2$ were subsequently sprayed to $FeO_x$, bimetallic SACs[36,37] (e.g., $Pd_1-Pt_1/FeO_x$, Fig. 3i–l) were obtained, which were confirmed by the AC HAADF-STEM image (the two metal species were atomically dispersed, Fig. 3i), element mapping analysis (Pd and Pt species were homogeneously distributed, Fig. 3j), and the EXAFS results (no Pd–Pd and Pt–Pt bond, Fig. 3k, l). The high flexibility of this approach for the combination of the catalytically active sites and underlying supports constructs a huge SACs matrix, which facilitates further exploration of practical applications.

## The large-scale production of $Pd_1/FeO_x$ SACs

Besides, the precursor-atomization strategy has great potentials for mass production of SACs. To fulfill the strategy, we designed a homemade production line (Fig. 4a, Supplementary Fig. 71, <$1000 in cost and <4 $m^2$ in floor area), which was composed of 34 ultrasonic atomizers, 16 infrared lamps, and 2 pieces of 2-m-long conveyor belts (to enable Step 1 (i.e., the spray of precursor solution onto the support surface) to be carried out continuously). The continuous synthesis process was shown in Supplementary Video 1. In addition, the productivity for $Pd_1/FeO_x$ discussed below can reach more than 1 kg $day^{-1}$ (Supplementary Fig. 72).

The supports, i.e., $FeO_x$, were spread out evenly (areal density: ~120 g $m^{-2}$) on the conveyor belt (running speed: ~1 cm $min^{-1}$), and it took about 6 h to complete the transfer (4 m long in total). Two conveyor belts were connected end to end, and the first conveyor belt was ~10 cm higher than the second one. As a result, $FeO_x$ was flipped over at the junction of the two conveyor belts and most of the surfaces of $FeO_x$ were exposed evenly. Similarly, the solution of $[Pd(NH_3)_4](NO_3)_2$ was atomized and sprayed at a rate of ~40 mL $h^{-1}$ onto $FeO_x$ on the conveyor belts, and the infrared lamps were employed to keep the $FeO_x$ powders dry (temperature > 60 °C). After being sprayed, the sample was collected at the end of the second conveyor belt and then calcined at 400 °C in a tube furnace. Four batches (every batch ~10 g) were collected at different times (each about 50 min apart, denoted as $Pd_1/FeO_x$-1, $Pd_1/FeO_x$-2, $Pd_1/FeO_x$-3, and $Pd_1/FeO_x$-4, respectively) were selected for the further investigation of catalyst structures and catalytic performances.

The four catalysts showed similar Pd loadings (~0.23 wt%, Supplementary Tables 47, 49–51) and specific surface areas (~44 $m^2 g^{-1}$, Supplementary Tables 47, 49–51) as well as a homogeneous distribution of Pd species (element mapping, Supplementary Figs. 73,

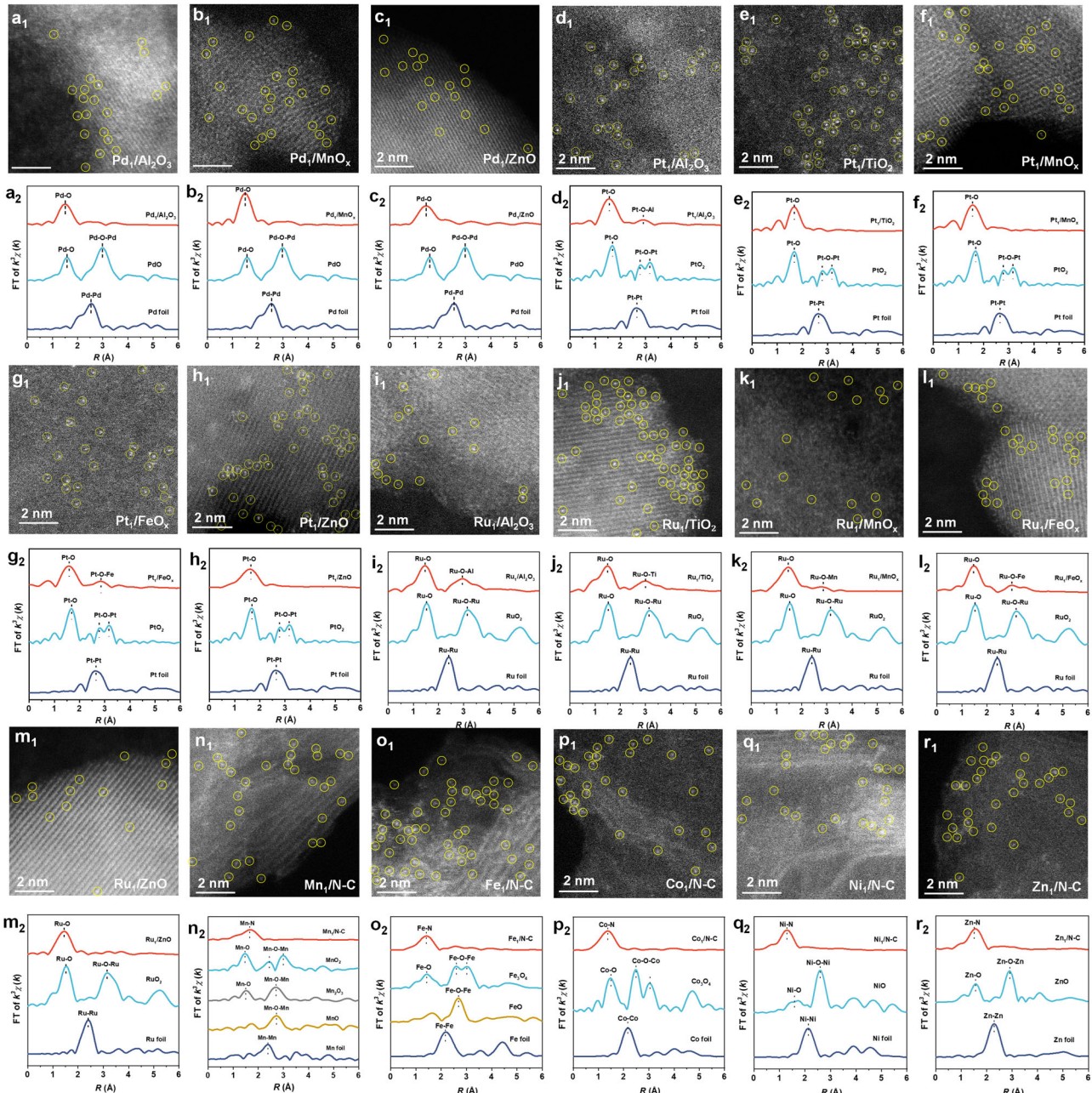

**Fig. 2 | The universality of the precursor-atomization strategy. $a_1$–$r_1$** AC HAADF-STEM images of $Pd_1/Al_2O_3$ ($a_1$), $Pd_1/MnO_x$ ($b_1$), $Pd_1/ZnO$ ($c_1$), $Pt_1/Al_2O_3$ ($d_1$), $Pt_1/TiO_2$ ($e_1$), $Pt_1/MnO_x$ ($f_1$), $Pt_1/FeO_x$ ($g_1$), $Pt_1/ZnO$ ($h_1$), $Ru_1/Al_2O_3$ ($i_1$), $Ru_1/TiO_2$ ($j_1$), $Ru_1/MnO_x$ ($k_1$), $Ru_1/FeO_x$ ($l_1$), $Ru_1/ZnO$ ($m_1$), $Mn_1/N$-C ($n_1$), $Fe_1/N$-C ($o_1$), $Co_1/N$-C ($p_1$), $Ni_1/N$-C ($q_1$), $Zn_1/N$-C ($r_1$). Scale bar, 2 nm. $a_2$–$r_2$ FT-EXAFS spectra of $Pd_1/Al_2O_3$ ($a_2$), $Pd_1/MnO_x$ ($b_2$), $Pd_1/ZnO$ ($c_2$), $Pt_1/Al_2O_3$ ($d_2$), $Pt_1/TiO_2$ ($e_2$), $Pt_1/MnO_x$ ($f_2$), $Pt_1/FeO_x$ ($g_2$), $Pt_1/ZnO$ ($h_2$), $Ru_1/Al_2O_3$ ($i_2$), $Ru_1/TiO_2$ ($j_2$), $Ru_1/MnO_x$ ($k_2$), $Ru_1/FeO_x$ ($l_2$), $Ru_1/ZnO$ ($m_2$), $Mn_1/N$-C ($n_2$), $Fe_1/N$-C ($o_2$), $Co_1/N$-C ($p_2$), $Ni_1/N$-C($q_2$), $Zn_1/N$-C ($r_2$).

75–77). Neither Pd/PdO characteristic signals in XRD patterns nor nanoparticles in TEM/STEM images were observed (Supplementary Figs. 73, 75–77). However, visible Pd atoms were present in AC HAADF-STEM images (Fig. 4b–e). Hence, it was deduced that the Pd species on all four catalysts were atomically dispersed. This was further supported by the very similar EXAFS and WT results for $Pd_1/FeO_x$-1and $Pd_1/FeO_x$-4 (Fig. 4f, g and Supplementary Figs. 74, 78; Supplementary Table 48, 52), including the coordination numbers (~4) and bond lengths (~2.0 Å) of Pd-O shell. The XANES spectra (Fig. 4h) suggested the positively charged Pd species on both $Pd_1/FeO_x$-1 and $Pd_1/FeO_x$-4. In addition, XPS curves for $Pd_1/FeO_x$-2, $Pd_1/FeO_x$-3, and $Pd_1/FeO_x$-4 resembled that of $Pd_1/FeO_x$-1 (Fig. 4i) and thus indicated the primary presence of $Pd^{2+}$ on all $Pd_1/FeO_x$ catalysts[38]. Clearly, the structures of Pd species on the four $Pd_1/FeO_x$

catalysts obtained at different time intervals were well retained, demonstrated by these complementary results above.

## The remarkable catalytic properties of Pd SACs
Suzuki–Miyaura cross-coupling is a typical homogeneous catalysis process for the formation of C–C bonds using soluble Pd/organo-phosphine ligands complex as catalysts, which entails tedious separation steps after reactions, thus leading to the increased cost and processing time[39,40]. To address the challenge and considering that SACs may combine the advantages of both homogeneous and heterogeneous catalysts, we tentatively tested the performance of 19 prepared SACs above with bromobenzene and phenylboronic acid as model substrates at 40 °C without inert gas protection (Fig. 5a, all the selectivities >99%). Pd-based SACs exhibited higher catalytic activity

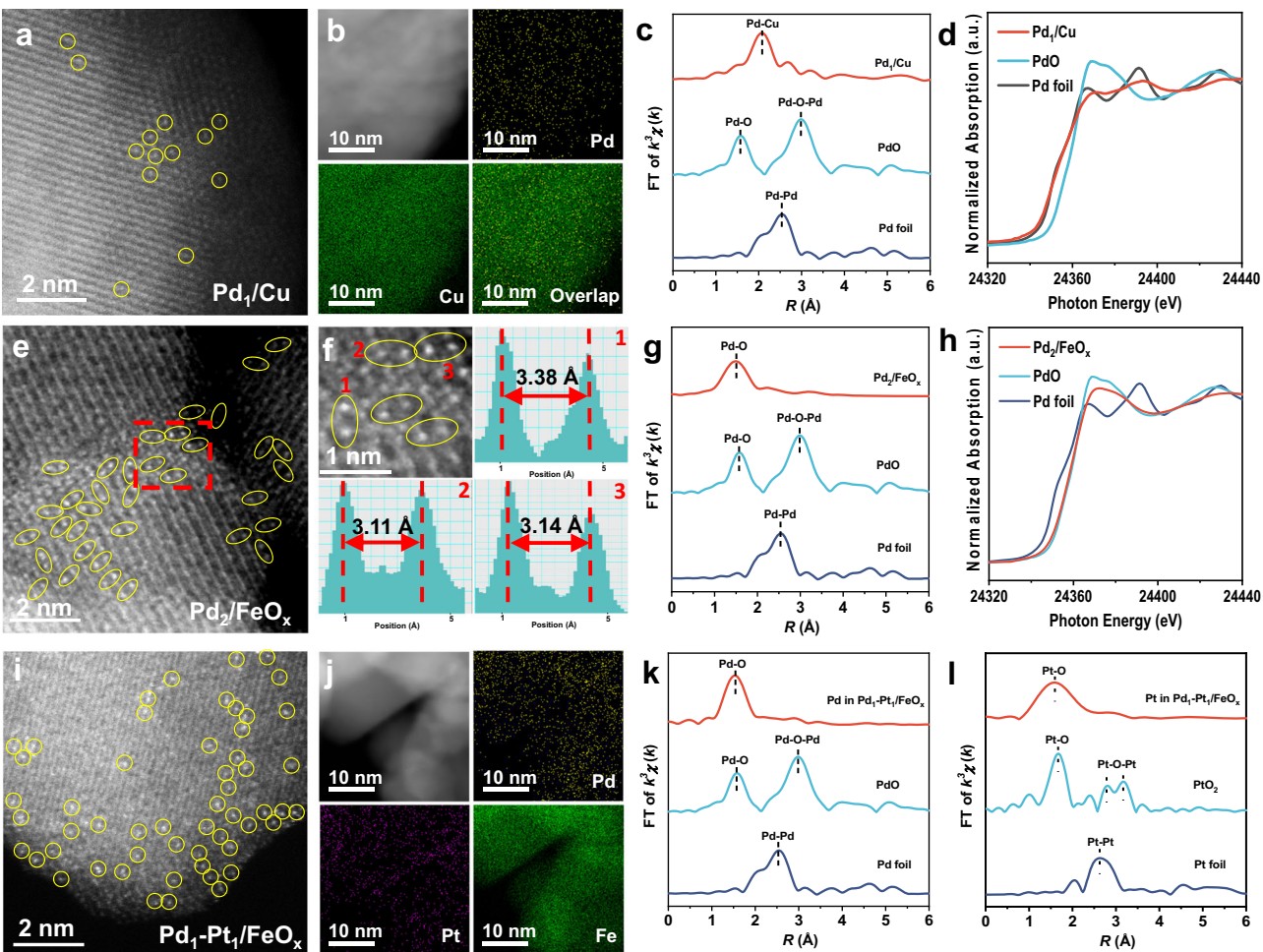

**Fig. 3 | Preparation and structural characterization of SACs derivatives. a–d** AC HAADF-STEM image, element mapping, FT-EXAFS, and XANES Pd K-edge spectra of $Pd_1/Cu$. **e–h** AC HAADF-STEM image, intensity profiles obtained in areas labeled 1–3, FT-EXAFS, and XANES spectra at Pd K-edge of $Pd_2/FeO_x$. **i–l** AC HAADF-STEM image, element mapping, and Pd K-edge FT-EXAFS spectra of $Pd_1$-$Pt_1/FeO_x$.

than other SACs (including Pt, Ru, and non-noble metals-based SACs), mainly because Pd was the most active metal species in the Suzuki–Miyaura cross-coupling reaction. In addition, $Pd_1/FeO_x$-1 showed higher activity than Pd SACs on other underlying supports. According to the previous reports[41], Suzuki–Miyaura cross-coupling reaction proceeds generally through a three-step pathway: oxidative addition, transmetalation, and reductive elimination, so the catalysts with superior redox properties are probably suitable for this reaction. In the $H_2$ temperature programmed reduction (TPR) results (Supplementary Fig. 79), $Pd_1/FeO_x$-1 showed the initial reduction peak at a lower temperature (-110 °C) than Pd SACs on other underlying supports, demonstrating the superior redox properties and accordingly the remarkable catalytic properties in Suzuki–Miyaura cross-coupling reaction. Moreover, the activity of the screened-out SACs of $Pd_1/FeO_x$-1 was much higher than that of commercial Pd catalysts, like $Pd(PPh_3)_4$, $Pd(NO_3)_2$, $Pd(OAc)_2$, $[Pd(NH_3)_4](NO_3)_2$, Pd/C, PdO, etc., as shown in Fig. 5b, which was also comparable with that of the top-tier heterogeneous catalysts under similar reaction conditions (Supplementary Table 53). To explore the ordinary compatibility of $Pd_1/FeO_x$-1 for the Suzuki–Miyaura cross-coupling reaction, the substrate scope was investigated using a variety of substituted aryl bromides and aryl boronic acids as raw materials. As shown in Fig. 6, under the optimized conditions, various combinations of aryl bromides with electron-donating and electron-withdrawing groups and aryl boronic acids with electron-donating and electron-withdrawing groups were successfully converted to the desired products, and excellent yields (93–99%) were successfully obtained without the protection of the inert atmosphere. The results thus suggested that $Pd_1/FeO_x$-1 exhibited a good substrate tolerance toward the Suzuki–Miyaura cross-coupling reaction.

Additionally, unlike the soluble catalysts (e.g., $Pd(PPh_3)_4$), $Pd_1/FeO_x$-1 can be easily separated by filtration, and no obvious activity loss was observed after 5 cycles (Fig. 5c). Notably, the AC HAADF-STEM image showed the catalysts maintained the atomically dispersed morphology after 5 catalytic runs (Supplementary Fig. 80) and the ICP results evidenced a negligible metal leaching in solutions. These results above indicated the superiority of $Pd_1/FeO_x$-1, including excellent catalytic properties, structural stability and recyclability. Remarkably, the catalysts of $Pd_1/FeO_x$-1, $Pd_1/FeO_x$-2, $Pd_1/FeO_x$-3, and $Pd_1/FeO_x$-4 exhibited very similar catalytic performance (see the small standard deviations for the TOFs of the four catalysts) under different reaction conditions, including temperatures ($9493 \pm 125$, $23810 \pm 1010$, and $51,944 \pm 1484 \, h^{-1}$ at 30, 40, and 50 °C, respectively, Fig. 5d), bases ($10,818 \pm 372$, $23,810 \pm 1010$, and $10,355 \pm 473 \, h^{-1}$ for $Na_2CO_3$, $K_2CO_3$, and $K_3PO_4$, respectively, Fig. 5e), or with different substrates ($5995 \pm 126$ and $111,573 \pm 4723 \, h^{-1}$ for aryl bromides bearing electron-donating and electron-withdrawing groups, respectively, Fig. 5f). Combined with the above results of structure characterizations, it demonstrated the identical catalytically active sites over the four $Pd_1/FeO_x$ catalysts and thus the excellent reproducibility of our synthesis approach, which was significant for the subsequent industrial applications.

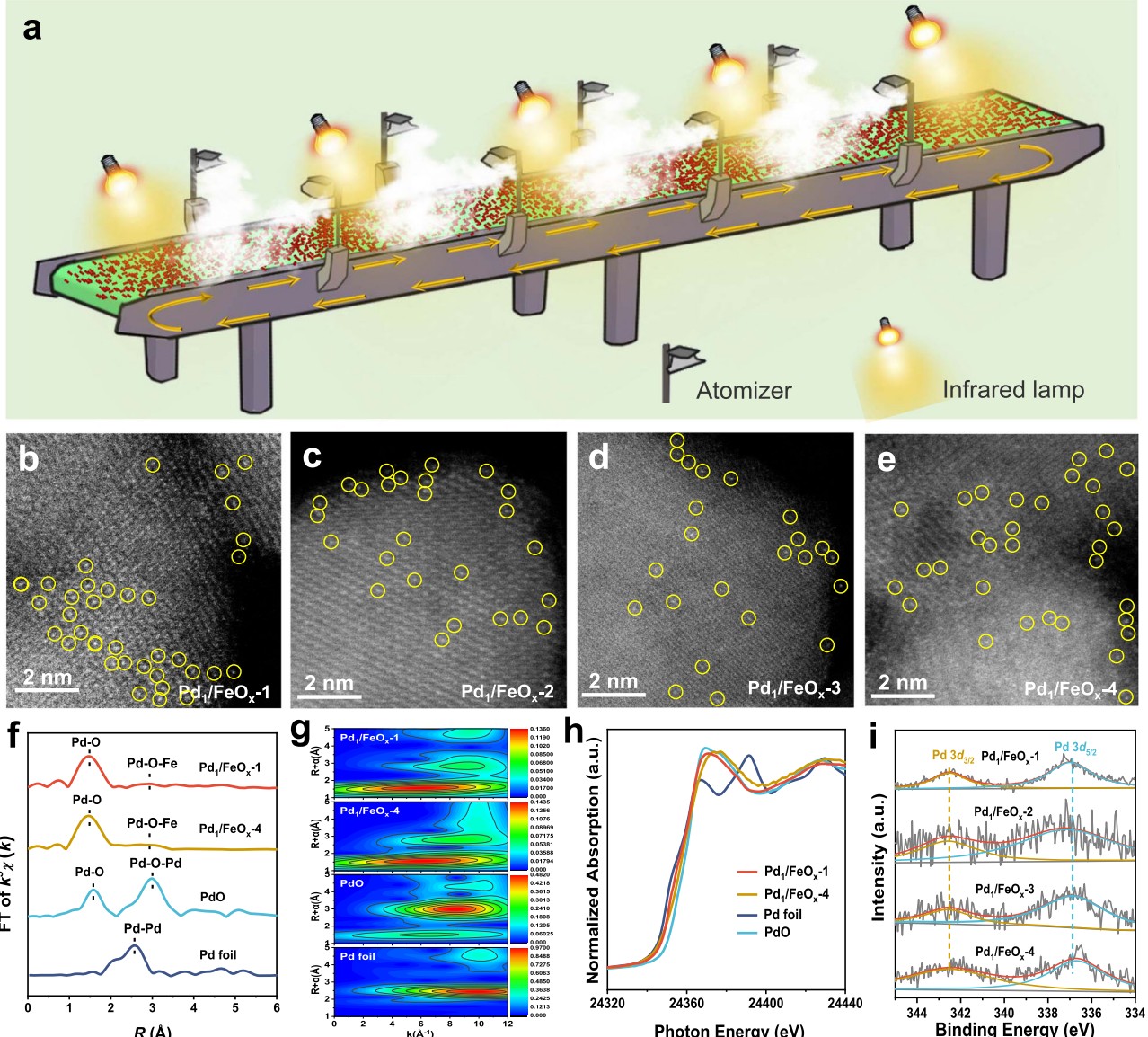

**Fig. 4 | The production line based on the precursor-atomization strategy. a** The scheme for the production line that is based on the precursor-atomization strategy. **b–e** AC HAADF-STEM image of $Pd_1/FeO_x$-1, 2, 3, 4, respectively. Pd atoms were highlighted by yellow circles. Scale bar, 2 nm. **f** The $k^3$-weighted FT spectra of Pd K-edge EXAFS for $Pd_1/FeO_x$-1 and 4, PdO, and Pd foil. **g** WT of Pd K-edge EXAFS of $Pd_1/FeO_x$-1, 4, PdO, and Pd foil. **h** Pd K-edge XANES for $Pd_1/FeO_x$-1 and 4, PdO, and Pd foil. **i** XPS patterns of Pd $3d$ for $Pd_1/FeO_x$-1, 2, 3, 4.

## Discussion

In summary, we have developed a versatile precursor-atomization strategy with great potential to precisely manipulate and arrange the atoms to obtain a library for SACs (including 19 SACs as well as SAAs, DACs, and bi-metallic SACs), basically covering the widely concerned catalyst structures of SACs at present. A homemade production line is established for the large-scale synthesis of SACs and the productivity can be increased to over 1 kg day$^{-1}$. The prepared $Pd_1/FeO_x$ shows identical catalyst structures and catalytic properties in Suzuki–Miyaura cross-coupling irrespective of different batches in production, demonstrating the remarkable reproductivity and thus the great potential for the large-scale synthesis of SACs.

## Methods
### Catalyst preparation

Here, we took $Pd_1/FeO_x$ as an example to illustrate the preparation process. The homemade equipment was shown in Supplementary Fig. 1, which was composed of an ultrasonic atomizer (Shenzhen Ikone Technology Co., Ltd), three infrared lamps (Royal Dutch Philips Electronics Ltd.), a heating plate, a plastic dome shield (ø 30 cm), an iron plate (ø 30 cm), and some cotton (for filling gap). First, take 12 g $FeO_x$ powders and spread them evenly on the iron plate. Then the tetraamminepalladium(II) nitrate aqueous solution (2.45 mmol/L) was sprayed onto the surface of $FeO_x$ through an atomizer with a rate of ~40 mL/h. To make the precursors deposit homogeneously onto the different surfaces of $FeO_x$, the $FeO_x$ powders were re-spread every hour. After a cumulative consumption of 0.16 L of solution, samples were collected and then treated at 400 °C (heating rate: 5 °C/min) for 2 h under 100 mL/min airflow in a tube furnace (Hefei Kejing Material Technology Co., Ltd), and $Pd_1/FeO_x$ (Pd: 0.27 wt%) was obtained. Afterward, the dome shield, iron dish, and cotton were thoroughly washed to recycle the unused precursors. Based on the ICP results, ~80% noble metal precursors ended up depositing on the supports, and ~7% on the plastic dome shield, the iron dish, and cotton, respectively, and the waste of precursors was <13%.

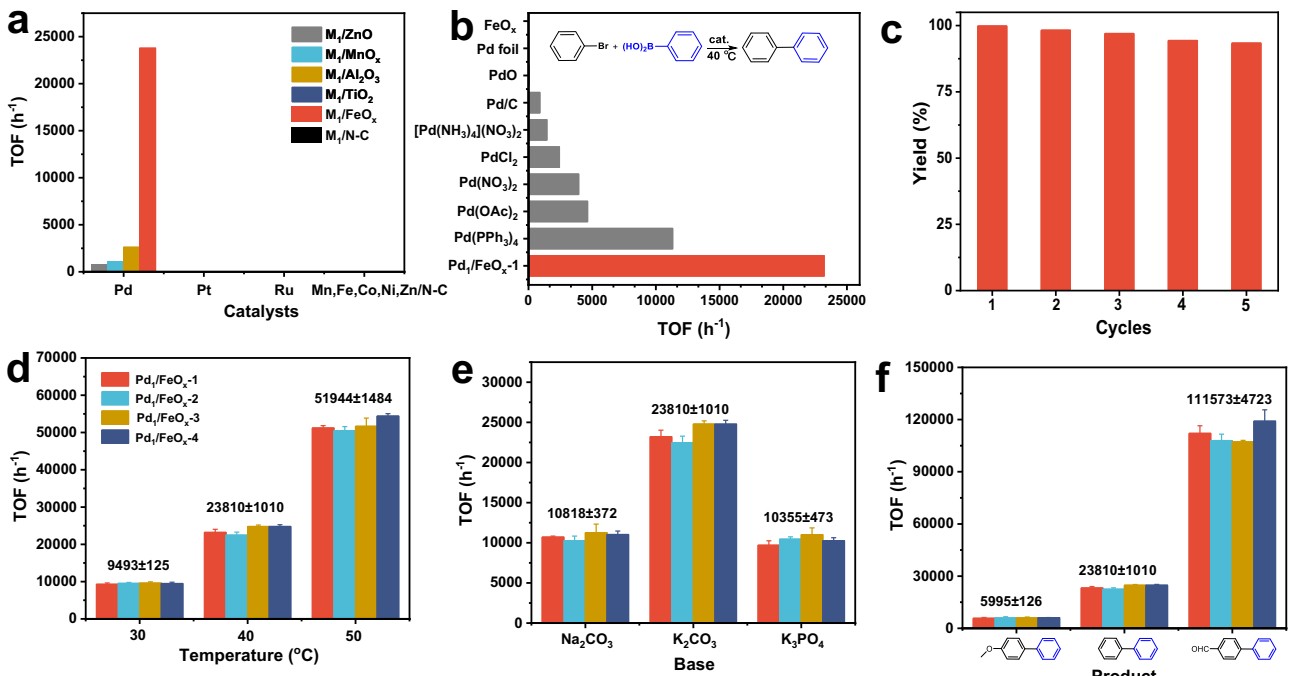

**Fig. 5 | Catalytic properties of Pd₁/FeOₓ in Suzuki–Miyaura cross-coupling reaction. a**, **b** Suzuki–Miyaura cross-coupling catalyzed by 19 SACs and different commercial Pd catalysts, respectively. Reaction conditions: 5 mg catalyst, bromobenzene (0.5 mmol), phenylboronic acid (0.75 mmol), K₂CO₃ (1.5 mmol), solvent (2 mL EtOH + 4 mL H₂O), 40 °C, decane as internal standard, biphenyl yields <30%. **c** Recycling experiments over Pd₁/FeOₓ-1. Reaction conditions were identical as **a**, except the amount of catalyst was 10 mg and the time was 2 h. **d** Catalytic performance of Pd₁/FeOₓ-1, 2, 3, 4 at different temperatures. Reaction conditions were

identical as **a**, except for the temperatures. **e** Catalytic performance of Pd₁/FeOₓ-1, 2, 3, 4 with different bases. Reaction conditions were identical as **a**, except for the bases. **f** Catalytic performance of Pd₁/FeOₓ-1, 2, 3, 4 for the formation of different products. Reaction conditions were identical as **a**, except for the substrates: 0.5 mmol (4-bromoanisole, bromobenzene or 4-bromobenzaldehyde) and 0.75 mmol phenylboronic acid, respectively. The error bars represent the standard deviation based on triplicate measurements.

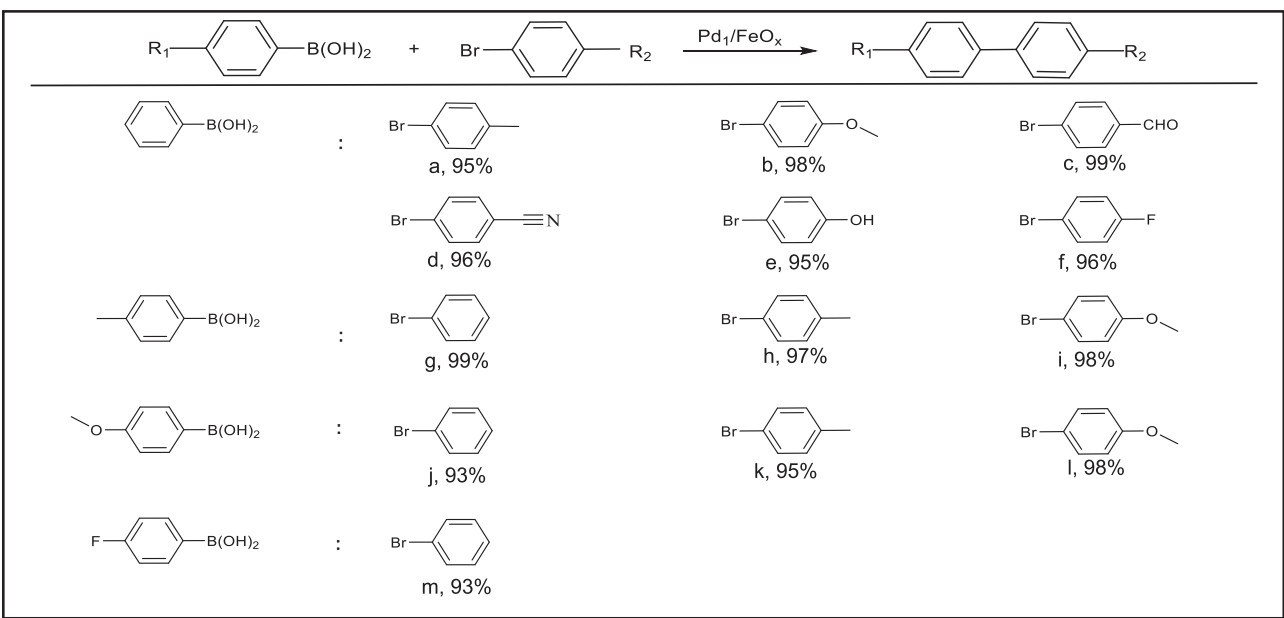

**Fig. 6 | Suzuki–Miyaura cross-coupling of aryl bromide and aryl boronic acids catalyzed by Pd₁/FeOₓ-1.** Reaction conditions: 10 mg catalyst, aryl bromide (0.5 mmol), aryl boronic acids (0.75 mmol), K₂CO₃ (1.5 mmol), solvent (2 mL EtOH + 4 mL H₂O), 40 °C, decane as internal standard, 2 h.

To verify the versatility of the precursor-atomization strategy, a series of catalysts, including Pd₁/Al₂O₃, Pd₁/MnOₓ, Pd₁/ZnO, Pt₁/Al₂O₃, Pt₁/TiO₂, Pt₁/MnOₓ, Pt₁/FeOₓ, Pt₁/ZnO, Ru₁/Al₂O₃, Ru₁/TiO₂, Ru₁/MnOₓ, Ru₁/FeOₓ, Ru₁/ZnO, Mn₁/N–C, Fe₁/N–C, Co₁/N–C, Ni₁/N–C, Zn₁/N–C, Pd₁/Cu, Pd₂/FeOₓ, and Pd₁–Pt₁/FeOₓ, were prepared with similar

synthesis procedures and the detailed parameters were given in Supplementary Table 3. For convenience, simplified homemade equipment was used here, which was only composed of an ultrasonic atomizer, an infrared lamp, a turntable (Shenzhen Xuriqiqi Technology Co., Ltd), and a glass dish. N–C was prepared as follows:

the mixture of carbon black and 2-methylimidazole (weight ratio is 1:1) was milled for 10 h at 400 r/min in a planetary ball mill (CHISHUN, QM3SP2).

The process of prepared $Pd_1/FeO_x$ on a large scale was as follows. The homemade production line was shown in Supplementary Fig. 71, which was composed of 2 pieces of 2-m-long conveyor belts (0.5-m width, Dongguan Hengyuntong Industrial Equipment Co., Ltd.), 34 ultrasonic atomizers, and 16 infrared lamps. The first conveyor belt was ~10 cm higher than the second one, and the atomizers and infrared lamps were evenly distributed on both sides of the conveyor belts. $FeO_x$ evenly dispersed (~120 g/m$^2$) on the surface of the conveyor belt was conveyed at a constant speed (~1 cm/min). Tetraamminepalladium(II) nitrate aqueous solution (2.45 mmol/L) was atomized and sprayed onto the $FeO_x$ through commercial ultrasonic atomizers (flow rate: ~40 mL/h). Afterward, the samples were collected at the end of the second conveyor belt and calcined in the tube furnace under airflow at 400 °C for 2 h.

## Characterization

EA was performed on a Vario EL cube instrument, and ICP-OES was carried out on a PerkinElmer OPTIMA 8000DV. A Micromeritic ASAP2020M analyzer was employed to conduct BET surface area measurements at liquid nitrogen temperature. The samples underwent vacuum treatment for about 6 h at 200 °C prior to the measurement. XRD measurements were carried out on a Bruker D8 Advanced diffractometer and the $2\theta$ range was 10–80°. XPS measurements were performed on an ESCALab250 XPS system equipped with an Al Kα source and a charge neutralizer, and all the binding energies were referenced to the contaminated C 1$s$ (284.8 eV). Both TEM and STEM images were obtained on an FEI Tecnai G2 F30, which was operated at 300 kV. AC HAADF-STEM images and element mapping were obtained on a JEM-ARM200F transmission electron microscopy incorporated with double spherical aberration correctors, operating at 200 kV. X-ray absorption spectroscopy (XAS) measurements were conducted on BL14W beamline at the Shanghai Synchrotron Radiation Facility (SSRF), 20-BM-B beamline at the Advanced Photon Source at Argonne National Laboratory (APS), and Singapore Synchrotron Light Source (SSLS). The samples were measured in fluorescence mode using Lytle detector (SSRF), 32-element Ge solid state detector (APS), or silicon drift detector (SSLS) and the corresponding metal foils and metal oxides were used as reference samples and measured in the transmission mode. Thermogravimetric analysis (TGA) results were obtained using a TG209F1 libra instrument. $^1$H and $^{13}$C nuclear magnetic resonance (NMR) spectra were obtained on Bruker Avance III 500 HD. H$_2$ TPR was performed on a DAS-7200.

## Catalytic performance test

The typical performance evaluation was carried out as follows: bromobenzene (0.5 mmol), phenylboronic acid (0.75 mmol), K$_2$CO$_3$ (1.5 mmol), 5 mg $Pd_1/FeO_x$ (Pd:substrate = 1:4000 (mol:mol)) and solvent (2 mL EtOH + 4 mL H$_2$O) were added to a 25 mL Schlenk tube, using decane as an internal standard. The mixture in the tube was stirred at 40 °C in the air atmosphere. After the reaction, the cooled mixture was first extracted with ethyl acetate (3 mL × 3), and the organic phase was collected and dried over anhydrous sodium sulfate. Product yields of the reaction were measured using GC (Shimadzu 2010 GC Plus), and the structure of the products was confirmed by GC–MS (Shimadzu GCMS-QP2010 Ultra) analysis. The corresponding NMR data are provided in Supplementary information (Supplementary Figs. 81–100). The catalyst was filtered and washed with ethanol and water several times, dried at 80 °C for 10 h and calcined in the air at 400 °C for 2 h, and then reused for the next cycle experiment.

The TOF is calculated at <30% conversion level and the Pd dispersion is assumed to be 100% (Eq. (1)):

$$TOF = n_0 * C * 106.42 / t * m_{cat} * \omega \quad (1)$$

where $n_0$ is the initial number of moles of the substrate, $C$ is the conversion of the substrate at reaction time $t$, $m_{cat}$ is the weight of the catalyst, and $\omega$ is Pd loading.

The yield is based on the bromobenzene or its derivatives, using decane as the internal standard, and the calculation equation is shown as Eq. (2).

$$Yield = n/n_0 \quad (2)$$

where $n$ is the number of moles of the product, and $n_0$ is the initial number of moles of the substrate.

The calculation equation of selectivity is shown as Eq. (3)

$$Selectivity = n/(n_0 * C) \quad (3)$$

where $n$ is the number of moles of product, $n_0$ is the initial number of moles of the substrate, and $C$ is the conversion of the substrate.

## Data availability

The data underlying Figs. 1–5, and Supplementary Figs. 2, 3, 5–60, 62–63, 65–70, 73–79 are provided as a Source Data file. The other data that support the findings of this study are available from the corresponding author upon request. Source data are provided with this paper.

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

## Acknowledgements

This work was supported by the National Key Research and Development Program Nanotechnology Specific Project (No. 2020YFA0210900, H.J.), National Key R&D Program of China (2021YFA1501102, D.M.), the Science and Technology Key Project of Guangdong Province, China (2020B010188002, H.J.), Guangdong Natural Science Funds for Distinguished Young Scholar (2022B1515020035, X.H.), Guangdong Provincial Key R&D Program (2019B110206002, H.J.), the National Natural Science Foundation of China (22078371, X.H.; 21938001, H.J.; 21961160741, H.J.; 22108315, Q.H.), Local Innovative and Research Teams Project of Guangdong Pearl River Talents Program (2017BT01C102, H.J.), the NSF of Guang-dong Province (2020A1515011141, X.H.), the Science and Technology Project of Guangzhou City, China (202102020461, H.H.).

## Author contributions

X.H. developed the concept, designed these experiments, and analyzed experimental data. X.H., H.Z., Y.C., and Y.Z. contributed to catalyst synthesis and catalytic experiments. H.Z., X.Z., and M.P. performed the EXAFS measurements and analyzed the data. X.H., H.Z., H.C., Q.H., X.Q., H.J., and D.M. wrote the paper. D.M. and H.J. directed the project. All authors discussed the results and commented on the manuscript.

## Competing interests

The authors declare no competing interests.
