## [Peer Review File · Nature Communications]

Title: Building up libraries and production line for single atom catalysts with precursor-atomization strategyREVIEWER COMMENTS

Reviewer #1 (Remarks to the Author):

In this article, Xiaohui He et al., reported the successful synthesis of a series of catalysts, including 19 SACs with different metal sites and supports and 3 derivatives of SACs via a versatile precursor-atomization strategy. This method involves the spraying precursor solution through ultrasonic atomization followed by the subsequent calcination, which can be scaled up via a homemade consecutive production line with productivity over 1 kg/day. The catalyst uniformity is further supported by the identical characterization results and catalytic properties in Suzuki-Miyaura cross-coupling. This finding is interesting. This work would make important contribution to the large-scale synthesis of SACs towards their industrial applications. Before the recommendation for publication in Nature Communications, the authors need to address the following questions.

1. The ultrasonic atomization pre-treatment is one of the key novel points in this paper. The authors need to provide more deep insights to justify why this step is crucial to achieve the generic synthesis. It is not fully convinced why ultrasonic atomization can address the different physical/chemical properties of precursor or supports that the authors listed in line 51-54. In addition, the authors need to highlight their major difference/conceptual advance as compared to previous work. For example, ultrasonic atomization coupled with a subsequent calcination has been reported for the mass production of SACs (Sci China Mater 2021, 64(3): 631–641).

2. The ligand removal process is a key step that may determine the configuration of SACs. The authors should provide more details. The thermogravimetric analysis in SI Fig 5 reveals that the precursor molecules can be decomposed between 200-250 °C. The author did not clarify why 400 °C was used in the calcination step.

3. What happens when Pd loading exceeds 0.27 wt%? any difference if one deposits 0.27 wt% Pd on FeOx using the conventional impregnation method? The authors need to comment on the upper limit of Pd loading using this approach.

4. The authors need to address the following issues related to the EXAFS characterization.

i. The R range of WT is very narrow.

ii. Authors need to show the FT-EXAFS of PdO in Fig. e, and WT-EXAFS of PdO K edge in fig. 4g.

iii. K range (and R range) for FT-EXAFS (and its fitting) should be specified.

iv. The authors focus on the discussion of the dominant peak at 1.6 Å in EXAFS spectra but the presence of sub-peak at 3.3 Å in the R space EXAFS Fig 3g is not explained. Similar issues can also be founded in other EXAFS spectra including Fig 1e, Fig 2I2, Fig 3g and spectra in SI section

v. The spectra as shown in Fig.2 don't support that all the samples exhibit the atomic-dispersion of metal atoms.

Some minor issues

- Line 359&396: 'The k3-weighted R-space FT spectra of EXAFS for...'

- o Authors should specify it is Pd K edge EXAFS.

- o R space EXAFS and FT-EXAFS are the exact same thing.

- Line 397: 'wavelet transform of...'. Specify it is WT of Pd K edge exafs.

- Line 102: 'only one notable peak at about 1.6 Å was shown...'. There is still a small peak around 3 Å for

sample Pd1/FeOx.

- Line 107: ‘...corresponding to the Pd–O–Fe bonding by comparing...’. What is Pd-O-Fe bonding?
 - Line 113: ‘...the energy absorption curve of Pd1/FeOx displayed a white line peak between...’.
 - o ‘Energy absorption curve’ is not a professional term. Please revise it.
 - Some of EXAFS fittings show very big error bar or not correct (such as the curve shown in SFig.29.3b, SFig.47.b, SFig.70.b). Improvement is necessary.
 - Some of XANES were displayed without energy calibration (such as the curve shown in SFig.31.a, SFig.34.a, SFig.37.a, SFig.40.a, SFig.43.a).
 - There is still a small peak around 3 Å for the Pd K edge FT-EXAFS of sample Pd1/FeOx-4.
 - Please indicate which samples were measured at SSRF, which samples were measured in aps, which samples were measured with lytle detector and which samples were measured with ssd detector.
5. Spraying Pd and Pt precursor molecules can yield bi-metallic SACs on FeOx. Do they occupy the same or different anchor sites? The authors need to clarify this point.
6. The synthesis of Pd-Au alloy is not fully convincing. It is well recognized that polycrystal Cu or Cu powders can be easily oxidized in air (Nature volume 603, pages434–438 (2022); Nature volume 586, pages390–394 (2020)). Here, there is a direct Pd-Cu bonding. Moreover, the Pd-Cu bonding length as revealed in R space of EXAFS in Fig 3c (about 2.1 Å) is not fully consistent with that shown in SI Fig 61. These issues need to be addressed.
7. More detailed information shall be provided to illustrate how the yield and TOF is calculated.
8. Provide the NMR data of the Suzuki-coupling products.
9. There are several typos (there are two panel g in Fig 2) and grammatical errors throughout this paper, (to name a few: This novel strategy lay a foundation (line 26); Pd-Pd bond was absence (line 104), The Pd:substrate is 1:40, mol%, not 1:4000 (line 315). the authors shall carefully amend the sentences.

Reviewer #2 (Remarks to the Author):

Ding Ma and coworkers reported the preparation of library of supported single atoms and evaluated their catalytic performance for the Suzuki-Miyaura cross-coupling. With respect to catalytic materials preparation and characterization, this work is quite interesting and strong enough. However, the catalytic application part is very weak and not up to the mark. Hence authors need to demonstrate the general applicability of their most active Pd-SACs with broad substrate scope by testing various simple, functionalized and challenging bromobenzenes and boronic acids. What about the reactivity of chlorobenzenes? Why Pd-SACs supported on iron oxide is found to be the best compared to other supported ones? What is the role of iron oxide support and why this support is more suitable compared to others? Authors have simply mentioned that ‘FeOx was the suitable support benefited from the surface basic sites’. However, there are other basic supports for example, MgO, ZnO, Al2O3 can also be the good supports. Why iron oxide is so special in this case? Compared to iron or metal oxides, other supports based on carbon and silica, are more abundant and inexpensive; why authors did not consider making more active Pd-SACs on these supports? It would be better to discuss in more detail about the role of support by appropriate characterization and mechanistic data. In the literature iron oxide

supported Pd-SACs are already reported for different catalytic applications including Suzuki–Miyaura and other cross coupling reactions. It is worthwhile to compare the activities of authors Pd-SACs catalyst with reported Pd-SACs and discuss how this catalyst is efficient and beneficial.

This work is interesting but needs suitable revision with additional work and results in order to make it more interesting and to attract the attention of scientists working in the area of catalysis and organic synthesis.

Response to the Reviewers' Comments:

Reviewer #1:

In this article, Xiaohui He et al., reported the successful synthesis of a series of catalysts, including 19 SACs with different metal sites and supports and 3 derivatives of SACs via a versatile precursor-atomization strategy. This method involves the spraying precursor solution through ultrasonic atomization followed by the subsequent calcination, which can be scaled up via a homemade consecutive production line with productivity over 1 kg/day. The catalyst uniformity is further supported by the identical characterization results and catalytic properties in Suzuki-Miyaura cross-coupling. This finding is interesting. This work would make important contribution to the large-scale synthesis of SACs towards their industrial applications. Before the recommendation for publication in Nature Communications, the authors need to address the following questions.

Response: We appreciate the reviewer's comment. And all the comments and suggestions were considered carefully and incorporated completely in the revision (see below for a point-to-point response).

Comment 1: The ultrasonic atomization pre-treatment is one of the key novel points in this paper. The authors need to provide more deep insights to justify why this step is crucial to achieve the generic synthesis. It is not fully convinced why ultrasonic atomization can address the different physical/chemical properties of precursor or supports that the authors listed in line 51-54. In addition, the authors need to highlight their major difference/conceptual advance as compared to previous work. For example, ultrasonic atomization coupled with a subsequent calcination has been reported for the mass production of SACs (Sci China Mater 2021, 64(3): 631–641).

Response 1: We thank the reviewer's comment. As discussed in manuscript, to understand the formation process of Pd₁/FeO_x, the sample (Pd₁/FeO_x-uncalcined), which was after being sprayed but before being calcined, were characterized. The visible Pd atoms in AC HAADF image (Fig. 1g in manuscript) and absence of Pd-Pd bonds in EXAFS results (Fig. 1e in manuscript) co-indicated that the Pd species were spatially isolated, demonstrating that the precursor-atomization strategy was able to make the precursors ([Pd(NH₃)₄](NO₃)₂) in the molecularly dispersed states. Then the samples above were calcined, and the molecularly dispersed [Pd(NH₃)₄](NO₃)₂ was

transformed into atomically dispersed Pd atoms (i.e., Pd₁/FeO_x was fabricated). In addition, we performed a controlled experiment. As shown in Fig. R1a, we used a pipette instead of an ultrasonic atomizer to spray the aqueous solution of tetraamminepalladium nitrate with the same concentration (2.45 mmol L⁻¹ [Pd(NH₃)₄](NO₃)₂) to the FeO_x supports. The drops produced by the pipette is about 2 μL (~2*10⁹ μm³ in volume), much larger than those produced by the ultrasonic atomizer (~40 μm³ in volume). In result, Pd nanoparticles with ~5 nm diameters were present in the FeO_x supports (Fig. R1b). It is presumed that the spray by the pipette is uneven and the moisture removal of the large drops is much slower than that of the small drops, possibly leading to the local aggregation of the Pd precursors and thus the appearance of Pd nanoparticles. The results above demonstrate the ultrasonic atomization pre-treatment play a crucial role in the preparation of SACs.

Furthermore, in the generic synthesis, the compatibility of the supports and the precursors should be taken into account. Here we give three examples.

First, the aqueous solution of tetraamminepalladium nitrate was atomized and sprayed to FeO_x, and after calcination the Pd₁/FeO_x SACs were obtained (Fig. R2a). The contact angle of FeO_x was measured to be 13.6 ° (Fig. R2b), indicative of its hydrophilicity. In addition, according to the literature, there are many anchor sites on FeO_x (Adv. Mater. 2022, DOI: 10.1002/adma.202201796), which help anchor the metal atoms firmly, avoiding the agglomeration during calcination.

Second, the aqueous solution of tetraamminepalladium nitrate was atomized and sprayed to carbon black, and after calcination the carbon black supported Pd nanoparticles appeared (~10 nm, Fig. R3a). The contact angle of carbon black was measured to be 141 ° (Fig. R3b), indicative of its hydrophobicity. Obviously, this is not conducive to the rapid diffusion of drops on the supports, which may result in the agglomeration of metal species caused by the high local concentration of precursors.

Third, the aqueous solution of tetraamminepalladium nitrate was atomized and sprayed to SiO₂, and after calcination the SiO₂ supported Pd nanoparticles also appeared (~5 nm, Fig. R4a). Different from carbon black, the contact angle of SiO₂ was measured to be only 7.2 ° (Fig. R4b), indicative of its hydrophilicity. So we speculate that the presence of Pd nanoparticles is mainly due to the weak anchor sites on SiO₂, which cannot anchor the metal atoms firmly to prevent the aggregation of Pd species during the calcination process (ACS Catal. 2022, 12, 2632).

Based on the results above, the basic prerequisites for the synthesis of SACs with the

precursor-atomization strategy are summarized: the compatibility of the supports and the solution of precursors, and the strong anchor sites on the supports. Luckily, the above requirements are not difficult to achieve, because most metal elements have water-soluble precursors and the commonly-used supports (e.g., FeO_x , Al_2O_3 , ZnO , TiO_2) are always hydrophilic and possess relatively strong anchor sites (especially under our relatively low calcination temperature (400 °C)). In consequence, if we take full account of the compatibility of the supports and the precursors, the general synthesis of SACs can probably be achieved (e.g., 19 SACs and 3 derivatives in this work). Further research (e.g., the combination of oil-soluble precursors and hydrophobic supports) is ongoing in our group.

In addition, there are obvious differences between the approach in the reference (denoted as Ma' method, *Sci. China Mater.* 2021, 64, 631) and ours. First, the preparation process is different. Ma's method includes at least 5 steps: 1) the KOH solution is nebulized and sprayed into the FeCl_2 solution to form $\text{Fe}(\text{OH})_x$ colloids; 2) the $\text{Fe}(\text{OH})_x$ colloids and phenanthroline are then mixed with carbon black (CB) under stirring; 3) the resulting mixture is pyrolyzed, forming FeNC SACs and some Fe nanoparticles (NPs) on the surface of the CB; 4) the product is then immersed in HCl to leach the Fe NPs; 5) another pyrolysis step is conducted to obtain final Fe SACs. Our method contains only two steps: 1) the dilute solution of the precursors is atomized and sprayed onto the supports; 2) the above samples undergo heat treatment to decompose the precursors, and the corresponding SACs are obtained. Obviously, our method is more facile. Second, the potentials for the mass production are different. Due to the complex process described above, Ma' method cannot realize the continuous production of SACs and the batch yield is only 5 g. In our method, coupled with conveyor belts, we are able to build a 1 kg/d production line. Based on the discussion above, benefited from the simple and straightforward preparation process, our method exhibits great potential for large-scale production of SACs.

Fig. R1 a) Photograph of the preparation process with a pipette instead of an ultrasonic atomizer. b) STEM image of Pd NPs/FeO_x.

Fig. R2 a) AC HAADF-STEM image of Pd₁/FeO_x. b) Contact angle measurement: photographs of a water droplet contacting with FeO_x.

Fig. R3 a) STEM image of Pd NPs/C. b) Contact angle measurement: photographs of a water droplet contacting with carbon black.

Fig. R4 a) STEM image of Pd NPs/SiO₂. b) Contact angle measurement: photographs of a water droplet contacting with SiO₂.

Comment 2: The ligand removal process is a key step that may determine the configuration of SACs. The authors should provide more details. The thermogravimetric analysis in SI Fig 5 reveals that the precursor molecules can be decomposed between 200-250 °C. The author did not clarify why 400 °C was used in the calcination step.

Response 2: We thank the reviewer's comment. Here we give more details. The samples after being sprayed were collected and put in a tube furnace (Hefei kejing material technology co., ltd). The samples were treated at 400 °C (heating rate: 5 °C /min) for 2 h under 100 mL/min air flow and taken out after naturally cooling to room temperature. As shown in Supplementary Fig. 5 in Supplementary information, the precursor molecules can be completely decomposed at about 270 °C, so the calcination temperature should be higher than 270 °C. Considering that too high calcination temperature may induce the aggregation of the metal species (Angew. Chem. Int. Ed. 2020, 59, 13562), the calcination temperature is set as 400 °C. In addition, 400 °C is also a commonly-used calcination temperature in the preparation process of Pd-based catalysts (Angew. Chem. Int. Ed. 2021, 60, 22769; J. Am. Chem. Soc. 2018, 140, 954).

Comment 3: What happens when Pd loading exceeds 0.27 wt%? any difference if one deposits 0.27 wt% Pd on FeOx using the conventional impregnation method? The authors need to comment on the upper limit of Pd loading using this approach.

Response 3: We thank the reviewer's comment. When Pd loading exceeded 0.27

wt% (e.g., 1.0 wt%), the surface density of Pd atoms increased (Fig. R5). However, when Pd loading was further increased to 1.5 wt% (Fig. R6), the Pd clusters or Pd nanoparticles emerged. In result, we think the upper limit of Pd loading on FeO_x supports using this approach is about 1 wt%.

Additionally, when we deposit 0.27 wt% Pd on FeO_x with the conventional impregnation method, due to the inevitable nonuniformity, some Pd nanoparticles are observed (~6 nm, Fig. R7), which clearly indicates the superiority of our method.

Fig. R5 AC HAADF-STEM image of Pd_1/FeO_x with 1 wt% Pd loading.

Fig. R6 AC HADFF STEM image of Pd/FeO_x with 1.5 wt% Pd loading.

Fig. R7 STEM image of Pd NPs /FeO_x prepared with conventional impregnation method.

Comment 4: The authors need to address the following issues related to the EXAFS characterization.

- i. The R range of WT is very narrow.
- ii. Authors need to show the FT-EXAFS of PdO in Fig. e, and WT-EXAFS of PdO K edge in fig. 4g.
- iii. K range (and R range) for FT-EXAFS (and its fitting) should be specified.
- iv. The authors focus on the discussion of the dominant peak at 1.6 Å in EXAFS spectra but the presence of sub-peak at 3.3 Å in the R space EXAFS Fig 3g is not explained. Similar issues can also be founded in other EXAFS spectra including Fig 1e, Fig 2I2, Fig 3g and spectra in SI section
- v. The spectra as shown in Fig.2 don't support that all the samples exhibit the atomic-dispersion of metal atoms.

Some minor issues

- Line 359&396: 'The k3-weighted R-space FT spectra of EXAFS for...'.
 - o Authors should specify it is Pd K edge EXAFS.
 - o R space EXAFS and FT-EXAFS are the exact same thing.
- Line 397: 'wavelet transform of...'. Specify it is WT of Pd K edge exafs.
- Line 102: 'only one notable peak at about 1.6 Å was shown...'. There is still a small peak around 3 Å for sample Pd₁/FeO_x.
- Line 107: '...corresponding to the Pd–O–Fe bonding by comparing...'. What is

Pd-O-Fe bonding?

- Line 113: ‘...the energy absorption curve of Pd₁/FeO_x displayed a white line peak between...’.
- o ‘Energy absorption curve’ is not a professional term. Please revise it.
- Some of EXAFS fittings show very big error bar or not correct (such as the curve shown in SFig.29.3b, SFig.47.b, SFig.70.b). Improvement is necessary.
- Some of XANES were displayed without energy calibration (such as the curve shown in SFig.31.a, SFig.34.a, SFig.37.a, SFig.40.a, SFig.43.a).
- There is still a small peak around 3 Å for the Pd K edge FT-EXAFS of sample Pd₁/FeO_x-4.
- Please indicate which samples were measured at SSRF, which samples were measured in aps, which samples were measured with lytle detector and which samples were measured with ssd detector.

Response 4: We thank the reviewer’s comment.

- i. The R range of WT was adjusted to 1 – 5 Å.
- ii. The FT-EXAFS of PdO was shown in Fig. R8 (i.e., Fig. 1e in manuscript), and the WT of Pd K edge EXAFS of PdO was shown in Fig. R9 (i.e., Fig. 4g in manuscript).

Fig. R8 The k³-weighted FT spectra of Pd K edge EXAFS for Pd₁/FeO_x, Pd₁/FeO_x-uncalcined, PdO, and Pd foil.

Fig. R9 WT of Pd K edge EXAFS of Pd₁/FeO_x-1 and 4, PdO and Pd foil.

iii. K range (and R range) for FT-EXAFS (and its fitting) was specified in Table R1.

Table R1 Data range for FT-EXAFS and its fitting

Sample	k range/Å ⁻¹	R range/Å
Pd ₁ /FeO _x	$2.8 \leq k \leq 8.7$	$1.0 \leq R \leq 4.3$
Pd ₁ /Al ₂ O ₃	$2.0 \leq k \leq 9.3$	$1.0 \leq R \leq 2.5$
Pd ₁ /MnO _x	$2.8 \leq k \leq 10.0$	$1.0 \leq R \leq 2.8$
Pd ₁ /ZnO	$2.0 \leq k \leq 8.0$	$1.0 \leq R \leq 3.0$
Pt ₁ /Al ₂ O ₃	$2.0 \leq k \leq 8.4$	$1.0 \leq R \leq 2.6$
Pt ₁ /TiO ₂	$3.0 \leq k \leq 10.5$	$1.3 \leq R \leq 2.7$
Pt ₁ /MnO _x	$3.1 \leq k \leq 9.3$	$1.3 \leq R \leq 4.0$
Pt ₁ /FeO _x	$3.5 \leq k \leq 9.0$	$1.1 \leq R \leq 4.0$
Pt ₁ /ZnO	$2.1 \leq k \leq 8.2$	$1.1 \leq R \leq 2.7$
Ru ₁ /Al ₂ O ₃	$2.8 \leq k \leq 9.2$	$1.2 \leq R \leq 2.7$
Ru ₁ /TiO ₂	$2.8 \leq k \leq 9.5$	$1.1 \leq R \leq 2.6$
Ru ₁ /MnO _x	$2.8 \leq k \leq 9.2$	$1.2 \leq R \leq 3.8$
Ru ₁ /FeO _x	$2.8 \leq k \leq 9.2$	$1.0 \leq R \leq 4.1$

Ru ₁ /ZnO	$2.7 \leq k \leq 10.4$	$1.1 \leq R \leq 2.6$
Mn ₁ /N-C	$2.1 \leq k \leq 9.7$	$1.0 \leq R \leq 2.2$
Fe ₁ /N-C	$2.5 \leq k \leq 8.9$	$1.0 \leq R \leq 3.6$
Co ₁ /N-C	$2.6 \leq k \leq 9.0$	$1.0 \leq R \leq 2.5$
Ni ₁ /N-C	$2.0 \leq k \leq 9.0$	$1.0 \leq R \leq 2.4$
Zn ₁ /N-C	$3.1 \leq k \leq 10.2$	$1.3 \leq R \leq 2.8$
Pd ₁ /Cu	$2.1 \leq k \leq 9.0$	$1.0 \leq R \leq 2.4$
Pd ₂ /FeO _x	$2.0 \leq k \leq 8.3$	$1.0 \leq R \leq 2.6$
Pd in Pd ₁ -Pt ₁ /FeO _x	$2.3 \leq k \leq 9.9$	$1.0 \leq R \leq 2.4$
Pt in Pd ₁ -Pt ₁ /FeO _x	$2.0 \leq k \leq 8.0$	$1.0 \leq R \leq 3.0$
Pd ₁ /FeO _{x-1}	$2.6 \leq k \leq 9.1$	$1.0 \leq R \leq 4.2$
Pd ₁ /FeO _{x-4}	$2.5 \leq k \leq 9.0$	$1.0 \leq R \leq 4.2$

iv. For Pd₁/FeO_x in Fig. 1e in manuscript.

There were two notable peaks at 1.5 and 2.9 Å (Fig. R10), which can be ascribed to the Pd-O and Pd-O-Fe scattering paths, respectively, in agreement with the previous report (Nat. Commun. 2020, 11, 3908). It should be noted that the peak at 2.9 Å cannot be ascribed to the Pd-Pd scattering path (Pd foil, 2.5 Å) and Pd-O-Pd scattering path (PdO, 3.0 Å), which was further confirmed by the adjusted EXAFS fitting results of Pd₁/FeO_x that these fitting results were in good agreement with the original curves (Fig. R11, i.e., Supplementary Fig. 4, and Table R2, i.e., Supplementary Table 2).

Fig. R10 The k^3 -weighted FT spectra of Pd K edge EXAFS for Pd_1/FeO_x , Pd_1/FeO_x -uncalcined, PdO, and Pd foil.

Fig. R11. a Fourier transform (FT) k^3 -weighted $\chi(k)$ -function of the EXAFS spectra for Pd K-edge and corresponding R-space fitting curves for the Pd_1/FeO_x catalyst. b EXAFS k space fitting curve and the experimental one of Pd_1/FeO_x .

Table R2 Structural parameters of EXAFS fitting for the Pd_1/FeO_x

Sample	Scattering pair	CN ^a	R (Å) ^b	$\sigma^2 (\times 10^{-3} \text{ \AA}^2)$ ^c	ΔE_0 (eV) ^d	R factor ^e
Pd_1/FeO_x	Pd-O	4.2 ± 0.6	2.01 ± 0.01	4.3 ± 1.7	1.0 ± 1.7	0.01
	Pd-O-Fe	0.8 ± 0.7	3.61 ± 0.02	4.7 ± 7.9		

For Ru₁/FeO_x in Fig. 2I₂ in manuscript.

There were two notable peaks at 1.5 and 3.0 Å (Fig. R12), which can be ascribed to the Ru-O and Ru-O-Fe scattering paths, respectively, in agreement with the previous report (JACS Au 2022, 2, 1078). It should be noted that the peak at 3.0 Å cannot be ascribed to the Ru-Ru scattering path (Ru foil, 2.4 Å) and Ru-O-Ru scattering path (RuO₂, 3.2 Å), which was further confirmed by the adjusted EXAFS fitting results of Ru₁/FeO_x that these fitting results were in good agreement with the original curves (Fig. R13, i.e., Supplementary Fig. 41, and Table R3, i.e., Supplementary Table 27).

Fig. R12 The k³-weighted FT spectra of Ru K edge EXAFS for Ru₁/FeO_x, RuO₂, and Ru foil.

Fig. R13 a FT k^3 -weighted $\chi(k)$ -function of the EXAFS spectra for Ru K-edge and corresponding R-space fitting curves for the Ru_1/FeO_x catalyst. b EXAFS k space fitting curve and the experimental one of Ru_1/FeO_x .

Table R3 Structural parameters of EXAFS fitting for the Ru_1/FeO_x

Sample	Scattering pair	CN ^a	R (Å) ^b	$\sigma^2 (\times 10^{-3} \text{ \AA}^2)$ ^c	ΔE_0 (eV) ^d	R factor ^e
Ru_1/FeO_x	Ru–O	4.3 ± 0.5	1.97 ± 0.01	1.0 ± 2.1	-1.4 ± 1.5	0.02
	Ru–O–Fe	0.8 ± 1.2	3.33 ± 0.03	3.8 ± 12.3		

For Pd_2/FeO_x in Fig. 3g.

The EXAFS data were carefully processed, and only the peak at 1.6 Å in Fig. R14, i.e., Fig. 3g in manuscript (R space EXAFS of Pd_2/FeO_x) were observed.

Fig. R14 The k^3 -weighted FT spectra of Pd K edge EXAFS for Pd_2/FeO_x , PdO, and Pd foil.

v. We carefully checked the data of each sample. Firstly, there was no M-M metal scattering path in R space, indicating that no metal nanoparticles or clusters were formed. Secondly, there was a strong M-O or M-N peak in the sample, and the position was significantly different from that of the M-O-M peak, indicating that no corresponding oxide was formed. Moreover, element mapping showed that all the elements in the samples were highly evenly distributed, and AC HAADF-STEM images showed that the target metal species were clearly distinguished in the support as single bright spots, which can be ascribed to the isolated metal atoms. All the complementary results above demonstrated that all the target SACs were successfully fabricated.

We thank the reviewer for pointing some minor issues. We have carefully checked the manuscript and corrected the issues.

We reprocess the EXAFS data and the corrected EXAFS fittings are shown in Fig. R15 (i.e., Supplementary Fig. 29b), Fig. R16 (i.e., Supplementary Fig. 47b), and Fig. R17 (i.e., Supplementary Fig. 70b).

Corrected XANES displayed with energy calibration are shown in Fig. R18 (i.e., Supplementary Fig. 31a), Fig. R19 (i.e., Supplementary Fig. 34a), Fig. R20 (i.e.,

Supplementary Fig. 37a), Fig. R21 (i.e., Supplementary Fig. 40a), Fig. R22 (i.e., Supplementary Fig. 43a).

Fig. R15 EXAFS k space fitting curve for Pt L₃-edge and the experimental one of Pt₁/ZnO.

Fig. R16 EXAFS k space fitting curve for Mn K-edge and the experimental one of Mn₁/N-C.

Fig. R17 EXAFS k space fitting curve for Pt L_3 -edge and the experimental one of Pt in $\text{Pd}_1\text{-Pt}_1/\text{FeO}_x$.

Fig. R18 XANES Ru K-edge for $\text{Ru}_1/\text{Al}_2\text{O}_3$, RuO_2 , and Ru foil.

Fig. R19 XANES Ru K-edge for Ru₁/TiO₂, RuO₂, and Ru foil.

Fig. R20 XANES Ru K-edge for Ru₁/MnO_x, RuO₂, and Ru foil.

Fig. R21 XANES Ru K-edge for Ru₁/FeO_x, RuO₂, and Ru foil.

Fig. R22 XANES Ru K-edge for Ru₁/ZnO, RuO₂, and Ru foil.

For Pd₁/FeO_x-1 in Fig. 4f in manuscript. There were two notable peaks at 1.5 and 2.9 Å (Fig. R23), which can be ascribed to the Pd-O and Pd-O-Fe scattering paths, respectively, in agreement with the previous report (Nat. Commun. 2020, 11, 3908). It should be noted that the peak at 2.9 Å cannot be ascribed to the Pd-Pd scattering path (Pd foil, 2.5 Å) and Pd-O-Pd scattering path (PdO, 3.0 Å), which was further confirmed by the adjusted EXAFS fitting results of Pd₁/FeO_x-1 that these fitting results were in good agreement with the original curves (Fig. R24, i.e., Supplementary Fig. 74, and Table R4, i.e., Supplementary Table 48). The above discussion also applies to Pd₁/FeO_x-4.

Fig. R23 The k^3 -weighted FT spectra of Pd K edge EXAFS for Pd_1/FeO_x -1, Pd_1/FeO_x -4, PdO, and Pd foil.

Fig. R24 a Fourier transform (FT) k^3 -weighted $\chi(k)$ -function of the EXAFS spectra for Pd K-edge and corresponding R-space fitting curves for the Pd_1/FeO_x -1 catalyst. b EXAFS k space fitting curve and the experimental one of Pd_1/FeO_x -1.

Table R4 Structural parameters of EXAFS fitting for the Pd₁/FeO_x-1

Sample	Scattering pair	CN ^a	R (Å) ^b	$\sigma^2 (\times 10^{-3} \text{ \AA}^2)$ ^c	ΔE_0 (eV) ^d	R factor ^e
Pd ₁ /FeO _x -1	Pd-O	4.0±0.5	2.04±0.01	2.6±1.3	-0.1±1.4	0.01
	Pd-O-Fe	0.2±0.4	3.39±0.02	24.9±14.2		

Fig. R25 a FT k^3 -weighted $\chi(k)$ -function of the EXAFS spectra for Pd K-edge and corresponding R-space fitting curves for the Pd₁/FeO_x-4 catalyst. b EXAFS k space fitting curve and the experimental one of Pd₁/FeO_x-4.

Table R5 Structural parameters of EXAFS fitting for the Pd₁/FeO_x-4

Sample	Scattering pair	CN ^a	R (Å) ^b	$\sigma^2 (\times 10^{-3} \text{ \AA}^2)$ ^c	ΔE_0 (eV) ^d	R factor ^e
Pd ₁ /FeO _x -4	Pd-O	3.7±0.7	2.01±0.01	1.2±2.0	-0.7±2.3	0.03
	Pd-O-Fe	1.5±2.4	3.42±0.03	6.3±14.3		

Detailed test source and detector for individual samples are shown in Table R6.

Table R6 Detailed test source and detector for individual samples

Sample	Source	Detector
Pd ₁ /FeO _x	SSLS	sdd
Pd ₁ /FeO _x -uncalcined	SSLS	sdd
Pd ₁ /Al ₂ O ₃	SSLS	sdd
Pd ₁ /MnO _x	aps	ssd
Pd ₁ /ZnO	SSRF	lytle
Pt ₁ /Al ₂ O ₃	SSLS	sdd
Pt ₁ /TiO ₂	SSLS	sdd
Pt ₁ /MnO _x	aps	ssd
Pt ₁ /FeO _x	aps	ssd
Pt ₁ /ZnO	SSRF	lytle
Ru ₁ /Al ₂ O ₃	SSLS	sdd
Ru ₁ /TiO ₂	SSLS	sdd
Ru ₁ /MnO _x	SSLS	sdd
Ru ₁ /FeO _x	SSLS	sdd
Ru ₁ /ZnO	SSLS	sdd
Mn ₁ /N-C	SSLS	sdd
Fe ₁ /N-C	SSLS	sdd
Co ₁ /N-C	SSLS	sdd
Ni ₁ /N-C	SSLS	sdd
Zn ₁ /N-C	aps	ssd
Pd ₁ /Cu	SSLS	sdd
Pd ₂ /FeO _x	SSLS	sdd
Pd in Pd ₁ -Pt ₁ /FeO _x	SSLS	sdd
Pt in Pd ₁ -Pt ₁ /FeO _x	SSRF	lytle
Pd ₁ /FeO _x -1	SSLS	sdd
Pd ₁ /FeO _x -4	SSLS	sdd

Comment 5: Spraying Pd and Pt precursor molecules can yield bi-metallic SACs on FeO_x. Do they occupy the same or different anchor sites? The authors need to clarify this point.

Response 5: We thank the reviewer's comment. First, in the AC HAADF-STEM image (Fig. 3i in manuscript), both the two metal species (Pd and Pt) were atomically dispersed. Second, the element mapping analysis (Fig. 3j in manuscript) indicated Pd and Pt species were homogeneously distributed. Third, the EXAFS results (Fig. 3k-l in manuscript) showed no Pd–Pd and Pt–Pt bond, and the fitting results showed that the coordination numbers of Pd–O and Pt–O (Supplementary Table 45 and 46 in Supplementary information) were very similar, i.e., 4.0 and 3.8, respectively. In addition, the characterization results of Pd₁/FeO_x did not show preferential location of Pd atoms on certain anchor sites (Fig. 1 in manuscript). So did Pt₁/FeO_x (Fig. 2g₁ and g₂ in manuscript). Based on the results above, we think possibly Pd and Pt randomly occupy the similar anchor sites.

Comment 6: The synthesis of Pd-Cu alloy is not fully convincing. It is well recognized that polycrystal Cu or Cu powders can be easily oxidized in air (Nature volume 603, pages434–438 (2022); Nature volume 586, pages390–394 (2020). Here, there is a direct Pd-Cu bonding. Moreover, the Pd-Cu bonding length as revealed in R space of EXAFS in Fig 3c (about 2.1 Å) is not fully consistent with that shown in SI Fig 61. These issues need to be addressed.

Response 6: We thank the reviewer's comment. First, the sizes of Cu particles used in this work were estimated to be about 60 nm by Scherrer Equation based on the XRD result (Fig. R26). According to the previous reports (J. alloy compd. 2018, 732, 240), the large Cu nanoparticles have certain antioxidant capacity. Second, in order to keep the Cu species in metallic state (evidenced by XRD results, Fig. R26), after being treated in 10% H₂/N₂ at 300 °C for 1 h, the samples were taken out after naturally cooling to room temperature. Then the samples were carefully kept in the glove box (Weige LG1200/750TS) at room temperature with low O₂ and H₂O contents (both <100 ppm), and used or characterized at a short time. Here we strictly avoid the harsh conditions that likely induce the oxidation of copper, like high temperature, salt spray, H₂O₂ treatment (Nature 2020, 586, 390) and long-term exposure to air (Nature 2022, 603, 434).

As to the EXAFS result of Pd₁/Cu, first, Pd-Cu scattering path as revealed in R space

of EXAFS in Fig. R27a (about 2.1 Å), different from those of Pd-O (1.56 Å), Pd-O-Pd (2.92 Å), and Pd-Pd (Pd foil, 2.46 Å). Second, after careful treatment, the wavelet transform (WT) plot (Fig. R27b) of Pd₁/Cu showed the maximum peak at (8.2 Å⁻¹, 2.3 Å), corresponding to the Pd-Cu scattering path by comparing Pd foil with the intensity maxima of Pd-Pd at (10.0 Å⁻¹, 2.5 Å), which agreed with the EXAFS results in R space (2.1 Å). Third, as shown in Fig. R27c, there were bright dots in the catalysts of Pd₁/Cu, which can be ascribed to the isolated Pd atoms, and the mapping result also indicated the homogenous distribution of Pd species on Cu (Fig. R27d). The complementary results above demonstrate the successful synthesis of Pd₁/Cu single atom alloys, also in line with the previous reports (ACS Catal. 2017, 7, 1491; Chem. Sci. 2019, 10, 8292).

Fig. R26 XRD of Pd₁/Cu.

Fig. R27 (a) Pd K-edge FT-EXAFS spectra of Pd₁/Cu, PdO, and Pd foil. (b) WT of Pd K-edge EXAFS of Pd₁/Cu and Pd foil. (c) AC HAADF-STEM image and (d) element mapping of Pd₁/Cu.

Comment 7: More detailed information shall be provided to illustrate how the yield and TOF is calculated.

Response 7: We thank the reviewer's comment.

The TOF is calculated at <30% conversion level and the Pd dispersion is assumed to be 100% (Equation R1):

$$\text{TOF} = n_0 \cdot C \cdot 106.42 / t \cdot m_{\text{cat}} \cdot \omega \quad \text{Equation R1}$$

where n_0 is the initial number of moles of substrate, C is the conversion of the substrate at reaction time t , m_{cat} is the weight of the catalyst, and ω is Pd loading.

The yield is based on the bromobenzene or its derivatives, using decane as internal standard, and the calculation equation is shown as Equation R2.

$$\text{Yield} = n / n_0 \quad \text{Equation R2}$$

where n is the number of moles of product, and n_0 is the initial number of moles of substrate.

Comment 8: Provide the NMR data of the Suzuki-coupling products.

Response 8: We thank the reviewer's comment. The corresponding NMR data are provided in Supplementary information (Supplementary Figs. 82-101).

Comment 9: There are several typos (there are two panel g in Fig 2) and grammatical errors throughout this paper, (to name a few: This novel strategy lay a foundation (line 26); Pd-Pd bond was absence (line 104), The Pd:substrate is 1:40, mol%, not 1:4000 (line 315). the authors shall carefully amend the sentences.

Response 9: We thank the reviewer's comment. The manuscript is carefully revised according to the suggestions.

Reviewer #2:

Ding Ma and coworkers reported the preparation of library of supported single atoms and evaluated their catalytic performance for the Suzuki-Miyaura cross-coupling. With respect to catalytic materials preparation and characterization, this work is quite interesting and strong enough. However, the catalytic application part is very weak and not up to the mark. ... This work is interesting but needs suitable revision with additional work and results in order to make it more interesting and to attract the attention of scientists working in the area of catalysis and organic synthesis.

Response: We appreciate the reviewer's comment. And all the comments and suggestions were considered carefully and incorporated completely in the revision (see below for a point-to-point response).

Comment 1: Hence authors need to demonstrate the general applicability of their most active Pd-SACs with broad substrate scope by testing various simple, functionalized and challenging bromobenzenes and boronic acids. What about the reactivity of chlorobenzenes?

Response 1: We thank the reviewer for the suggestion.

To explore the ordinary compatibility of Pd₁/FeO_x for the Suzuki–Miyaura cross-coupling reaction, the substrate scope was investigated using a variety of substituted aryl bromides and aryl boronic acids as raw materials. As shown in Fig. R28, under the optimized conditions, various combinations of aryl bromides with electron-donating and -withdrawing groups and aryl boronic acids with electron-donating and -withdrawing groups were successfully converted to the desired products, and excellent yields (93~99 %) were successfully obtained without the protection of the inert atmosphere. The results thus suggested that Pd₁/FeO_x exhibited a good substrate-tolerance toward the Suzuki–Miyaura cross-coupling reaction.

However, when chlorobenzenes were subjected to the reaction, the yields were only 4-5% (Fig. R28g and n). Higher temperature and prolonged reaction time barely improved the catalytic performances to 8% yield (Fig. R28g), possibly owing to the strong coordination of Cl⁻ ions and Pd atoms, which may poison the active sites of Pd₁/FeO_x catalyst (Small 2020, 16, 2001782).

Fig. R28 Suzuki-Miyaura cross-coupling of aryl bromide and aryl boronic acids catalyzed by Pd₁/FeO_x. Reaction conditions: 10 mg catalyst, aryl bromide (0.5 mmol), aryl boronic acids (0.75 mmol), K₂CO₃ (1.5 mmol), solvent (2 mL EtOH + 4 mL H₂O), 40 °C, decane as internal standard, 2 h. I. 60 °C, 4 h.

Comment 2: Why Pd-SACs supported on iron oxide is found to be the best compared to other supported ones? What is the role of iron oxide support and why this support is more suitable compared to others? Authors have simply mentioned that 'FeO_x was the suitable support benefited from the surface basic sites'. However, there are other basic supports for example, MgO, ZnO, Al₂O₃ can also be the good supports. Why iron oxide is so special in this case? Compared to iron or metal oxides, other supports based on carbon and silica, are more abundant and inexpensive; why authors did not consider making more active Pd-SACs on these supports? It would be better to discuss in more detail about the role of support by appropriate characterization and mechanistic data. In the literature iron oxide supported Pd-SACs are already reported for different catalytic applications including Suzuki-Miyaura and other cross coupling reactions. It is worthwhile to compare the activities of authors Pd-SACs catalyst with reported Pd-SACs and discuss how this catalyst is efficient

and beneficial.

Response 2: We thank the reviewer's comment.

First, we attempt to prepare Pd SACs on different underlying supports as mentioned. Pd₁/FeO_x, Pd₁/MgO, Pd₁/ZnO, and Pd₁/Al₂O₃ were obtained (Fig. R29). However, due to the inert surface of carbon and silica (Nat. Nanotechnol., 2022, 17, 606; ACS Catal., 2022, 12, 2632), Pd species on carbon and silica tended to form nanoparticles (Fig. R30) instead of Pd SACs with our precursor-atomization strategy.

Second, according to the previous reports (Chem. Rev. 2018, 118, 2249; Angew. Chem. Int. Ed. 2013, 52, 7362), Suzuki coupling reaction proceeds generally through a three-step pathway: oxidative addition (dissociative adsorption of arylhalide), transmetalation (dissociative adsorption of alkali-activated phenylboronic acid), and reductive elimination (coupling of two phenyl groups). The catalyst is required to serve as a charge donor in the oxidative addition step and a charge acceptor in both transmetalation and reductive elimination steps of the reaction (J. Catal. 2018, 360, 20). Accordingly, the catalyst with superior redox properties may be suitable for this reaction (Small 2020, 16, 2001782). H₂-TPR measurements for Pd₁/FeO_x, Pd₁/MgO, Pd₁/ZnO, and Pd₁/Al₂O₃ were performed (Fig. R31). As expected, only Pd₁/FeO_x showed low temperature reduction peak (~100 °C), clearly indicating the good redox properties of Pd₁/FeO_x. Considering that all Pd species of the four SACs were atomically dispersed, the good redox properties of Pd₁/FeO_x can be ascribed to the underlying support of FeO_x, which also lead to the superior catalytic performance of Pd₁/FeO_x in the Suzuki-Miyaura cross-coupling reactions. The corresponding revision is made in the manuscript.

Furthermore, the previously reported catalytic systems for Suzuki–Miyaura cross-coupling reaction were listed for comparison (Table R7). Obviously, under similar reaction conditions, our Pd₁/FeO_x SACs provided higher or comparable TOFs, possibly because of the high metal dispersion and superior redox properties.

Fig. R29 AC HADFF STEM image of (a) Pd₁/FeO_x, (b) Pd₁/MgO, (c) Pd₁/ZnO, and (d) Pd₁/Al₂O₃.

Fig. R30 STEM image of (a) Pd NPs/C and (b) Pd NPs/SiO₂.

Fig. R31 H₂-TPR measurements for Pd₁/FeO_x, Pd₁/MgO, Pd₁/ZnO, and Pd₁/Al₂O₃. Measurement conditions: 30 mg catalyst, 30 mL/min 5% H₂/N₂, room temperature to 500 °C with 10 °C/min rate.

Table R7. The catalytic activities in Suzuki–Miyaura cross-coupling reaction of Pd₁/FeO_x with other reported FeO_x-supported Pd catalysts

Entry	Catalyst	Solvent	Temp. (°C)	TOF (h ⁻¹)	Ref.
1	Pd ₁ /FeO _x	EtOH-H ₂ O	30	9493	This work
2	Pd ₁ /FeO _x	EtOH-H ₂ O	40	23810	This work
3	Pd ₁ /FeO _x	EtOH-H ₂ O	50	51944	This work
4	Pd(0/II)/CS-biguanid@Fe ₃ O ₄	EtOH-H ₂ O	RT	240	Int. J. Biol. Macromol. 2018, 113, 186
5	Fe ₃ O ₄ /IL/Pd	EtOH-H ₂ O	RT	475	Tetrahedron Lett. 2017, 58, 4269
6	Pd/Fe ₃ O ₄ NPs	EtOH-H ₂ O	RT	960	Appl. Organomet. Chem. 2020, 34, e5653
7	Fe ₃ O ₄ @Pectin/Pd	EtOH-H ₂ O	RT	1280	Int. J. Biol. Macromol. 2020, 160, 1252
8	Fe ₃ O ₄ @SiO ₂ -TCT-GA-Pd(0)	EtOH-H ₂ O	50	533	Appl. Organomet. Chem. 2021, 35,

9	Fe ₃ O ₄ /CS-Me@Pd	EtOH-H ₂ O	50	74000	Int. J. Biol. Macromol. 2021, 184, 358
10	Fe ₃ O ₄ @AOFC/Pd(II)	EtOH-H ₂ O	55	4615	Carbohydr. Polym. 2017, 177, 165
11	H-Fe ₃ O ₄ @TiO ₂ -NH ₂ /Pd	EtOH-H ₂ O	60	287	J. Chin. Chem. Soc. 2018, 65, 875
12	Fe ₃ O ₄ dpa@Pd0.5	H ₂ O	65	3150	RSC Adv. 2016, 6, 68675
13	Fe ₃ O ₄ @SiO ₂ -EDTA-Pd NPs	EtOH-H ₂ O	70	194	Appl. Organomet. Chem. 2018, 32, e4302
14	Fe ₃ O ₄ @PFC-Pd(0)	DES (K ₂ CO ₃ /glycerol)	70	522	Carbohydr. Polym. 2020, 235, 115947
15	Pd-γ-Fe ₂ O ₃	H ₂ O	80	128	J. Organomet. Chem. 2018, 871, 96
16	Fe ₂ O ₃ @CSF@Pd	DMF-H ₂ O	80	513	Mol. Catal. 2020, 493, 111042
17	Fe ₃ O ₄ @SiO ₂ @NHC@Pd-MNPs	i -PrOH-H ₂ O	80	37691	J. Organomet. Chem. 2021, 943, 121823
18	Fe ₃ O ₄ @SiO ₂ @N-amidinoglycine@Pd0	H ₂ O	90	3388	Transit. Metal Chem. 2018, 43, 295

REVIEWERS' COMMENTS

Reviewer #1 (Remarks to the Author):

The authors have addressed all the questions, and thus I recommend the publication of this work in the current form.

Reviewer #2 (Remarks to the Author):

Authors have well addressed the comments and revised the manuscript and SI. However, I suggest including all the catalytic reaction results (for example, Supplementary Fig. 80) into the main manuscript and to discuss the results. The paper should be balanced and interesting to read for both materials and synthetic research community.

Response to Referees' Comments

We thank the referees for their valuable comments and positive endorsement to our manuscript. We have carefully considered the referees' comments and revised the manuscript. Our responses and corresponding revisions are as follows:

Response to the Reviewers' Comments:

Reviewer #1:

The authors have addressed all the questions, and thus I recommend the publication of this work in the current form.

Response: We appreciate the reviewer's comment.

Reviewer #2:

Authors have well addressed the comments and revised the manuscript and SI. However, I suggest including all the catalytic reaction results (for example, Supplementary Fig. 80) into the main manuscript and to discuss the results. The paper should be balanced and interesting to read for both materials and synthetic research community.

Response: We appreciate the reviewer's comment. According to the suggestion, Fig. 6 (i.e., Supplementary Fig. 80 in the last version) is included in the main manuscript.